# Groundwater Recharge Potentiality Mapping in Wadi Qena, Eastern Desert Basins of Egypt for Sustainable Agriculture Base Using Geomatics Approaches

Hanaa A. Megahed [1,*], Abd El-Hay A. Farrag [2], Amira A. Mohamed [2], Paola D'Antonio [3], Antonio Scopa [3,*] and Mohamed A. E. AbdelRahman [4,5]

1 Division of Geological Applications and Mineral Resources, National Authority for Remote Sensing and Space Sciences (NARSS), Cairo 1564, Egypt

2 Geology Department, Faculty of Science, Assiut University, Assiut 71526, Egypt; elhay829@aun.edu.eg (A.E.-H.A.F.); amira.abdelraheem@scince.aun.edu.eg (A.A.M.)

3 Scuola di Scienze Agrarie, Forestali, Alimentari ed Ambientali (SAFE), Università degli Studi della Basilicata, Viale dell'Ateneo Lucano, 10, 85100 Potenza, Italy; paola.dantonio@unibas.it

4 Division of Environmental Studies and Land Use, National Authority for Remote Sensing and Space Sciences (NARSS), Cairo 1564, Egypt; maekaoud@gmail.com or maekaoud@narss.sci.eg

5 Key Laboratory of Agricultural Remote Sensing, Ministry of Agriculture and Rural Affairs, Institute of Agricultural Resources and Regional Planning, Chinese Academy of Agricultural Sciences, Beijing 100081, China

* Correspondence: hanaa.ahmed@narss.sci.eg (H.A.M.); antonio.scopa@unibas.it (A.S.); Tel.: +20-10-1503-5625 (H.A.M.)

**Abstract:** In arid and hyper-arid areas, groundwater is a precious and rare resource. The need for water supply has grown over the past few decades as a result of population growth, urbanization, and agricultural endeavors. This research aims to locate groundwater recharge potential zones (GWPZs) using multi-criteria evaluation (MCE) in the Wadi Qena Basin, Eastern Desert of Egypt, which represents one of the most promising valleys on which the government depends for land reclamations and developments. These approaches have been used to integrate and delineate the locations of high groundwater recharge and the potential of the Quaternary aquifer in the Wadi Qena basin. After allocating weight factors to identify features in each case based on infiltration, land use/land cover, slope, geology, topology, soil, drainage density, lineament density, rainfall, flow accumulation, and flow direction, these thematic maps were combined. The results of the GIS modeling led to the division of the area's groundwater recharge potential into five groups, ranging from very high (in the western part) to very low (in the eastern part of the basin). The zones with the best prospects for groundwater exploration turned out to be the alluvial and flood plains, with their thick strata of sand and gravel. The groundwater recharge potential map was validated using data from the field and earlier investigations. The promising recharging areas show high suitability for soil cultivation. The results overall reveal that RS and GIS methodologies offer insightful instruments for more precise assessment, planning, and monitoring of water resources in arid regions and anywhere with similar setups for groundwater prospecting and management.

**Keywords:** groundwater; remote sensing; potentiality; GIS; Wadi Qena basin; sustainable agriculture; Egypt

## 1. Introduction

Egypt, a country with limited water resources, has experienced a rapid increase in water demand due to population growth, resulting in it falling below the water poverty line. Over the past two centuries, Egypt's per capita share of water has decreased from approximately 2000 m$^3$/year to around 600 m$^3$/year (https://www.capmas.gov.eg; accessed on 22 August 2023). This decline has made Egypt one of the poorest countries in the world, as the water poverty limit is around 1000 m$^3$/year. As a result, Egypt



has become increasingly reliant on groundwater to meet its water needs, as surface water alone is insufficient throughout the year. To address the population problem and expand arable land, the Egyptian government has announced plans to invest in desert reclamation. However, the success of this endeavor in the Wadi Qena Basin, located in Egypt's eastern desert, depends on the availability of groundwater. The Wadi Qena basin receives an estimated average annual rainfall of $1.4 \times 10^8$ m$^3$ [1,2]. In the Middle East and Northern Africa regions, wadis, which are dry ravines or gullies, are commonly found. The term "wadi" originates from the Arabic language and translates to "valley" or "ravine". While wadis can experience a sudden surge of water during the rainy season, they typically remain dry or have sporadic stream flow for the majority of the year.

Aquifer recharge zones are globally identified through a combination of remote sensing (RS) and Geographic Information System (GIS) techniques, as demonstrated by numerous studies [3–8]. These studies effectively pinpoint the most favorable areas for aquifer recharge, taking into account similarities with the research region [5,9–20]. The identification of groundwater prospecting zones is primarily based on the integration of multiple criteria, including topography, stream networks, slope steepness, lithology, and the frequency of lineaments, as suggested by some researchers. To evaluate these criteria simultaneously, the widely used method of multi-criteria evaluation (MCE) is commonly employed [3,21–27]. The outcomes of this approach provide compelling evidence regarding the condition of the aquifer [28]. In the context of Malaysia's Kedah Peninsula, researchers have utilized the analytical hierarchical process (AHP) to delineate groundwater potential zones (GWPZs) [29].

The utilization of AHP approaches by the authors of [30–33] allowed for an exploration of GWPZs and recharge rates. The AHP approach simplifies complex outcomes into a series of pairwise data, resulting in the desired outcomes [34]. AHP is widely recognized as a highly effective technique for evaluating output consistency, mitigating bias, and implementing it within a novel context [30]. The integration of various factors, such as geomorphology, geology, lineament density, slope, drainage density, and rainfall, within the GIS model is crucial for assessing aquifer potential and identifying recharge zones. This comprehensive approach incorporates a weightage calculation using AHP, enabling a more accurate evaluation of the aquifer's potential and the identification of areas with high recharge capacity. In Egypt, previous studies have made use of hydrogeological and geological thematic maps in combination with RS and GIS techniques. These maps have been employed by numerous Egyptian researchers to investigate various aspects related to the geology and hydrology of the region [2,35–40]. Gheith and Sultan [35] developed a hydrological model to forecast the rate at which sporadic precipitation in the Red Sea hills would replenish alluvial aquifers in the Eastern Desert. By analyzing and interpreting optical/microwave data and conducting fieldwork, Abdelkareem [36] successfully predicted areas of groundwater accumulation in Wadi Qena using geomatics, RS, and GIS environments.

In the central part of the primary Wadi Qena basin, there exists the entrance to six sub-basins. Abdel Moneim [37] conducted a study on hydrogeology to assess the potential of groundwater for sustainable development. The investigation initially focused on the potential for flood hazards, followed by an examination of the properties and infiltration capacity of the uppermost layer of soil. Lastly, an evaluation of the quantity and quality of groundwater was conducted. Geomorphometric analyses indicated that rapid runoff with limited chances of downward infiltration was expected. The infiltration test results revealed that 70% of the tests exhibited infiltration rates below 2 m/day, with an average rate of 1.97 m/day. This could be attributed to the prevalence of silt, loam, and clayey soils in the region. Consequently, this highlights the limited recharge of groundwater and the significant loss of water through runoff. The research concluded that a majority of the local groundwater could be utilized for irrigating crops that can tolerate medium to high levels of salt. However, it is not suitable for private household use.

Moneim et al. [38] conducted a study on the groundwater in Wadi Qena Basin, which originates from the Quaternary and Nubian aquifers. The study involved a hydrogeological analysis and radiocarbon dating to determine the age of the groundwater. The primary objective of the study was to assess the suitability of the groundwater for human consumption and agricultural use. The thickness of the Quaternary aquifers increases downstream of a wadi, ranging from 20 to 100 m. On the other hand, the Nubian aquifer has an average thickness of 350 m and is 300–500 m deep, with decreasing depth towards the north. Abd El-Hameed et al. [39] conducted hydrogeological and geoelectrical investigations on the Wadi Qena Basin, which led to the identification of two distinct aquifer systems: the shallow aquifer and the Nubian aquifer. The Nubian aquifer is found in the southern part of the Wadi and is characterized by confined conditions, whereas in the northern part, it exists in unconfined conditions. The groundwater in Wadi Qena primarily consists of fossil water, with the Nubian aquifer being minimally recharged by rainfall and locally by the dissecting faults of the eastern and western plateaus that intersect the Wadi. It is not anticipated that the basement complexes will serve as a significant source of recharge for the aquifer system. Hussien et al. [2] employed a multidisciplinary methodology to map the potential of groundwater in the Wadi Qena basin. The study utilized various techniques such as RS, geophysics, isotopes, stratigraphy, structural interpretation, and geochemistry to sample the Quaternary aquifer, Nubian aquifer system, post-Nubian aquifer system, and fracture crystalline aquifer. The authors recommended that groundwater extractions in the Wadi Qena basin should be located near the identified, deeply buried faults based on the hydrogeological history of the region. In another study, an integrated approach was used to characterize the groundwater resources in the southern portion of Wadi Qena Basin. The approach included hydrogeochemical analysis, remote sensing, GIS, and field data [40]. The study utilized five thematic layers from satellite images and field data, namely topography, lineament, lithology, slope, and stream network, to create a groundwater prospective map. The final integrated maps depicted areas with different groundwater potential and quality. The study concluded that the region's groundwater is only suitable for irrigation of crops that are highly salt-tolerant.

As a result of the construction of a new road connecting upper Egypt governorates and the Red Sea, various forms of agricultural development have advanced in the Wadi. In the current study, an assessment software package (Arc GIS 10.3) using STRM (DEM) data with 30 m resolution was used to prepare topography, slope, aspect, contour, drainage network, flow accumulation, and flow direction maps of the Wadi Qena Basin. Several satellite images, including Shuttle Radar Topography Mission (STRM) and Landsat data, were processed and combined to predict the best groundwater recharge potentiality zones. GIS spatial analysis was used to combine several raster layers of topography, geology, slope drainage density, soil, rainfall data, and lineament maps. This method successfully identifies the best groundwater recharge potential locations. Using the GIS reclassify tool, the results potential map was classified into five zones, ranging from very high to very low groundwater recharge potential zones, which is an essential contribution to water management in Wadi Qena Basin. Drilled wells were used to confirm the accuracy of the final map, which showed a number of underutilized groundwater-capable locations. Planners and decision-makers can utilize the groundwater potential zone (GWPZ) map to guide future development. Meanwhile, it is an important input for sustainable agriculture use.

Initially, early interpretations of land focused on its agricultural potential, taking into account factors such as productivity ratings and soil suitability for specific crops [41–45]. However, the emergence of sustainable farming practices has led to an increased importance of soil information in GIS. This has resulted in the development of customized variable-rate application plans that cater to the unique conditions of individual fields, with soil-landscape information derived from soil interpretations becoming a crucial tool for effective land management [46,47]. In many cases, legacy soil data are utilized to visualize, analyze, and model the connections between soil properties and hydrologic processes, as well as the relationship between soil variability and hydropedologic properties [48,49].

Soil water assessment is the most commonly employed soil input parameter when assessing soil, encompassing various factors such as soil texture class, depth to bedrock, bulk density, porosity, permeability, available water capacity, soil-hydrologic group, and curve numbers [50,51]. The soil data used in this study were obtained through field surveys and soil analyses, providing a comprehensive dataset that includes both physical and chemical properties of the soil. Given the circumstances, it is important to acknowledge that the inputs utilized in this study to evaluate soil suitability for crop cultivation remain consistent. Additionally, recognizing the significance of water availability in ensuring the long-term sustainability of agricultural soil, all supplementary soil studies were thoroughly examined. Consequently, during the fieldwork, both indicators for assessing soil suitability and water recharge were thoroughly investigated.

This study endeavors to ascertain the spatial allocation of shallow aquifers that are conducive for habitation, agricultural activities, and industrial progress. This will be achieved through the comprehensive assessment of aquifer-influencing parameters, identification of areas where aquifer recharge occurs, and ultimately, the development of a GWPZ map utilizing MCE techniques.

## 2. Materials and Methods

### 2.1. Description of the Study Area

Wadi Qena Basin is in the Eastern Egyptian Desert, specifically between the coordinates of 26°10′ and 28°05′ North, and 32°31′ and 32°45′ East (Figure 1). The area of Wadi Qena Watershed is 16,000 km². The deltaic mouth of Wadi Qena Basin is located in the city of Qena, and its course extends northward at latitude 28°05′ N. The Wadi itself is a main channel that runs through the floor of a larger valley. The limestone plateau cliffs form its western boundary, while the Red Sea mountains frame its eastern edge.

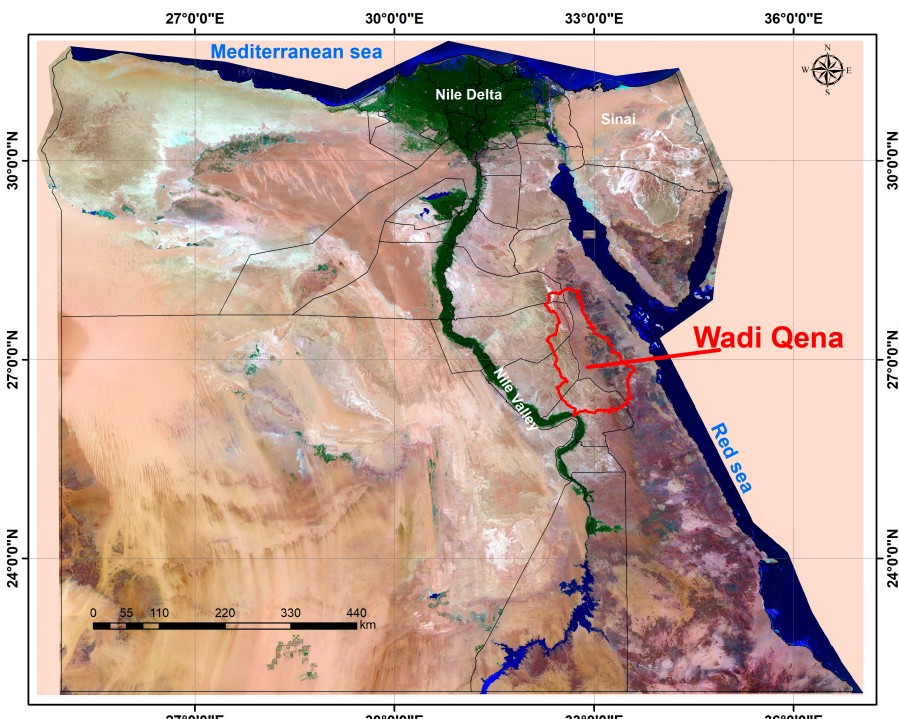

**Figure 1.** Location map of Wadi Qena Basin.

### 2.2. Geological and Hydrological Setting

To the south, amidst the Red Sea peaks and the valley, lies a vast expanse characterized by its wide and flat terrain. This expansive plain is adorned with distinctive brown tabular formations composed of Nubia sandstone, which are overlain by subsequent sedimentary layers [52].

One of the Eastern Desert's most interesting basins is Wadi Qena Basin. There are numerous studies and publications on the Wadi Qena Basin geological sequence, including those by [52–62]. Different geological formations occupy the main channel of Wadi Qena Basin on its eastern and western sides. The igneous and metamorphic rocks represented in the Red Sea mountains cover the eastern areas. The remaining areas of the Wadi are composed of Paleozoic sedimentary rocks, mostly in the center and south. The Nubian Formation overlies basement rocks with thicknesses ranging from 14 to 403 m [60]. The geological formation of the Upper Cretaceous is situated above the Nubian Formation, which is unconformably covered by Paleocene, Eocene, and Pliocene rocks, forming the western calcareous plateau and downhill area. The Quaternary deposits on the surface are observed in the shape of alluvial hills and terraces, composed of gravels, sands, and fine clay materials [39]. The rock units (Figure 2a) of Wadi Qena Basin can be ordered from top to bottom as follows: Holocene, Pleistocene, Pliocene sediments, Phanerozoic sedimentary sequence and basement rocks (Pre-Cambrian) [36,53–55,63–65].

Hydrogeologically, many authors including [2,28,37,38] have summarized the groundwater aquifers system in the Wadi Qena Basin area. According to [38], the Wadi Qena Basin's hydrologic units can be divided into five water-bearing horizons: Quaternary, shale or clay, fracture limestone, Nubian sandstone, and basement or fractured crystalline aquifer (Figure 2b). The Quaternary aquifer is located at the top of the hydrogeological section and consists of various types of deposits such as gravels, conglomerate, and sandstone with silt and sand interbedded, with thicknesses ranging from a few meters to more than 75 m. In the north, the thickness decreases, and the facies change to clay, shale, and silt. These sediments are thought to be a shallow aquifer. Shale or clay is considered as aquiclude horizon; it consists of impervious deposits such as clay and shale with a thickness ranging from 50 to more than 100 m.

The fractured limestone and dolomite formation, which belongs to the Upper Cretaceous age, is primarily composed of fractured limestone and dolomite with intercalations and lenses of marl and gypsum. This aquifer has a thickness ranging from 100 to 250 m and is considered moderately productive, containing fresh to brackish water. The Nubian sandstone aquifer, which is the main sandstone aquifer in the area, has a thickness ranging from 100 to 350 m and is composed of sandstone with clay intercalations. Finally, the basement complex consists of various types of fractured igneous and metamorphic rocks, forming a fractured crystalline aquifer. Conduits in the form of cracks and joints play a crucial role in recharging the aquifers in the Wadi region. The rocks in this area are classified as an aquifuge horizon, with a thickness exceeding 2000 m. The Quaternary and Nubian aquifers are the most promising sources of groundwater. Groundwater movement is predominantly in the NE–SW direction, with the Nubian aquifer in the Wadi Qena Basin flowing from northeast to southeast. The thickness of the Quaternary aquifer of alluvial deposits varies due to effective erosion of surface layers and undulation of subsurface strata. Downstream of the Wadi, the thickness of the Quaternary aquifer increases until it reaches 100 m. The Nubian aquifer's depth varies depending on its location, and is determined by geological conditions. On average, it is situated 300 to 500 m below the ground [38]. This aquifer serves as the primary water source for the residents of the Wadi Qena Basin. In the research area, the Quaternary aquifer is unconfined, and the water depths range from 2.70 m (msl) to 35.95 m (msl) from the ground surface. The aquifer's water table map (Figure 2c) indicates a recharge area located in the northeastern region of the study area. This recharge is attributed to the upward flow from the Nubia sandstone aquifer through NW–SE faults and fractured basement rocks, as well as surface recharge from the upstream courses of alluvium deposits in the wadies. As a result, the groundwater flow direction in this aquifer is classified as northeast to southwest. However, excessive pumping for irrigation activities in the southern part of the study area has led to a second flow direction from north to south. The estimated discharge from shallow dug wells and open pits reaches 107 m$^3$/h, and the total drawdown after 4 h of pumping reaches 1.22 m [66].

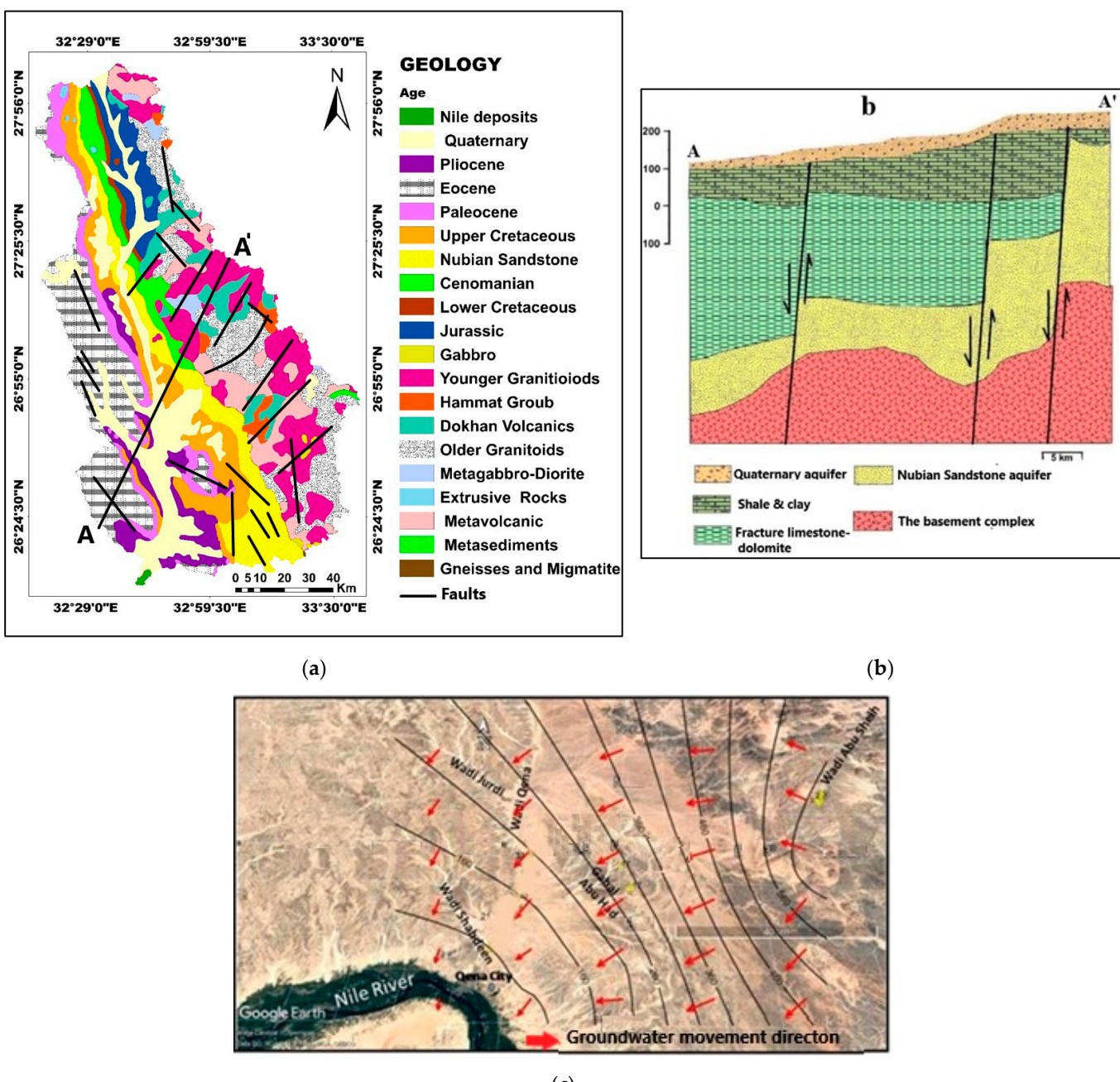

**Figure 2.** (**a**) Geological map of Wadi Qena After EGSMA, [63] and UNESCO [67]. (**b**) Hydrogeological cross-section in Wadi Qena Abd El Moniem [37]. (**c**) Water table map for the Quaternary alluvium aquifer at Wadi Qena basin, Eastern Desert Egypt.

*2.3. The Approach to Generating Groundwater Recharge Potential Maps*

In this paper, we present a cost- and time-effective and interdisciplinary research study for the creation of GWPZ maps. At first, we collected and reviewed the related information from the space and ancillary data (geological and hydrogeological previous studies), as shown in Figure 3. Furthermore, the climatic data from NASA and various meteorological stations located in Assuit, Hurghaga, Kos-sir, Luxor, Minya, Sohag, and South Valley University have been collected and analyzed for a span of 30 years. This meticulous approach serves as a deliberate means of obtaining comprehensive information regarding the climatic conditions in the valley. Also, some Landsat satellite images were used, including Landsat thematic mapper (TM) images (1990) and Sentinel-2 mages (2023)

were used for studying the land cover and land use change. The valley's rock structures were made clear using band math and the band ratio equation. To accomplish this task, the software tools employed are ENVI 5.3 and ARC GIS 10.3. A Digital Elevation Model (DEM) is used for terrain analysis and hydrogeological through ARC GIS 10.3. Also, morphometric parameters of basins in the study area were extracted using ArcMap and Excel for mathematical calculations. Using a raster calculator and ArcMap 10.3, the morphometric parameters (topography, slope, soil, lithology/geology, drainage density, lineaments, land use/land cover data, rainfall, flow accumulation, and flow direction) were used to identify the high-potential recharge areas and prospect the valley's potential high-groundwater locations. The potential of groundwater resources has been mapped out through the integration of remote sensing and GIS data using the weighted overlay method.

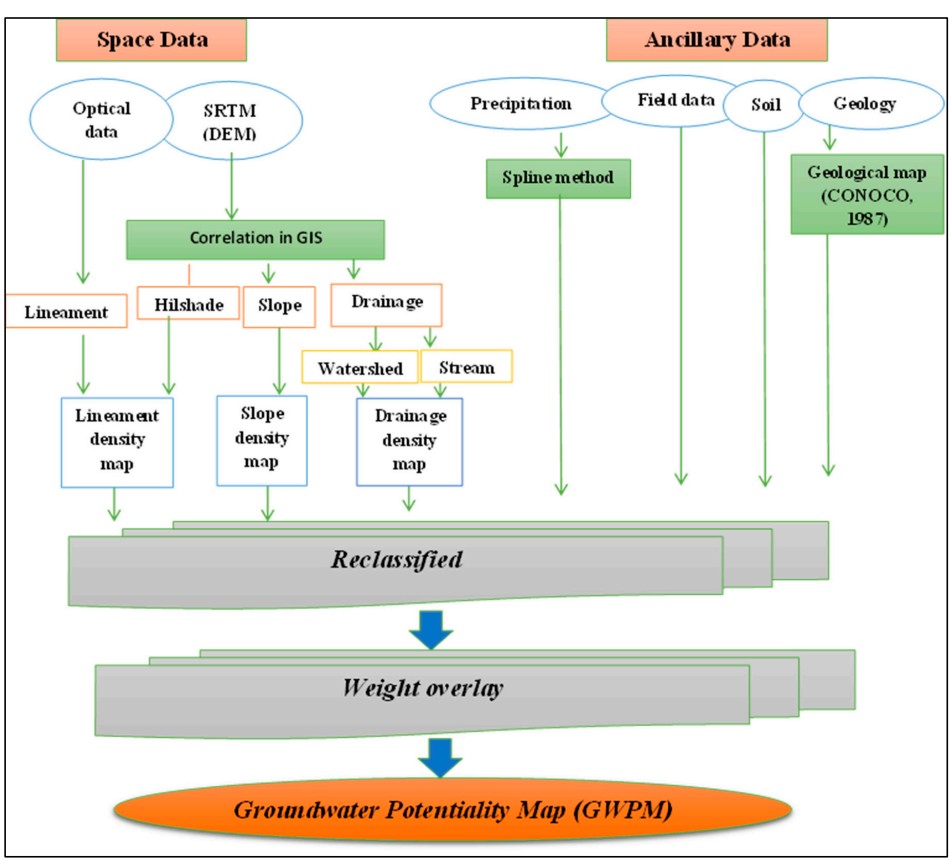

**Figure 3.** Flow chart displaying the approach to generating groundwater recharge potential maps.

Each thematic layer, such as DEM, geology–lithology, drainage density, slope, topography, soil, rainfall, lineaments and land use/land cover, flow accumulation, and flow direction maps were given evidence for groundwater conditions. The GIS layer has been created by converting the rank of each layer, which is determined by dividing the total sum of the layer ranks by the rank of each map. The capability values (CVi) are generated by dividing the rank of each class by the total summation values of the layer classes. The groundwater potential zone map is calculated by multiplying these capability values with the respective layer weight in each thematic map. The possible occurrences are then aggregated in a linear combination equation using the Raster Calculator as a function of the Spatial Analyst tool in ArcMap GIS. This integration process is carried out as per the given instructions.

$$GWP = \sum CVi \times Wi \qquad (1)$$

where GWP = groundwater potential, CVi = capability value (weight of inter-map class), Wi = map weight.

Emerging study areas in the fields of groundwater hydrology, resource management, environmental monitoring, and emergency response involve the integration of remote sensing (RS), geographic information systems (GIS), and the global positioning system (GPS). Recent developments in the disciplines of RS, GIS, GPS, and higher-level computation will aid in the timely and economically efficient provision and handling of a variety of data. This flowchart (Figure 3) explains hydrological modeling, how remote sensing and GIS are used, the requirement for integration models, and finally, its conclusion. After addressing these problems philosophically and technically, we will be better equipped to handle big data quantities and disseminate information about fast-diminishing societal resources.

## 3. Results

### 3.1. Remote Sensing and GIS to Determine Groundwater Potential Zones

The hydrogeological conditions affect how much groundwater is available, so the topography, slope, soil, lithology/geology, drainage density, lineaments, land use/land cover data, rainfall, flow accumulation, and flow direction were all considered to determine where groundwater collects in Wadi Qena Basin. Each layer was put into a category based on how important it was to control how groundwater came in or out. Based on what had been mentioned before [36,68], the weights of the different layers were calculated. By putting these layers together with Arc GIS software (10.3), a map was made that showed where groundwater could be found in the Wadi Qena Basin.

### 3.2. Lithology/Geological Map

Infiltration of surface water, runoff, and recharging of aquifers are all affected by how porous and permeable sediments are. Based on the geological map of Egypt (UNESCO, 2005) and a Landsat 8 image, the lithology of Wadi Qena Basin was classified into four groups based on their relative infiltration tests and some previous studies by [27,40,69,70] on similar rock units include, basement rocks, Nile deposits, Pliocene-Pleistocene and Wadi deposits, which are assigned the numbers 1, 2, 3, and 4, respectively. Number 1 shows low infiltration and high runoff, while number 4 shows high infiltration, which is what the Wadi deposits show (Figure 4a). Wadi deposits primarily comprise of substantial alluvium, unconsolidated gravel, sand, and silt, characterized by significant intergranular spaces that facilitate the effortless percolation of water into the underlying layers. Consequently, these deposits exhibit a remarkably favorable capacity for recharging groundwater resources.

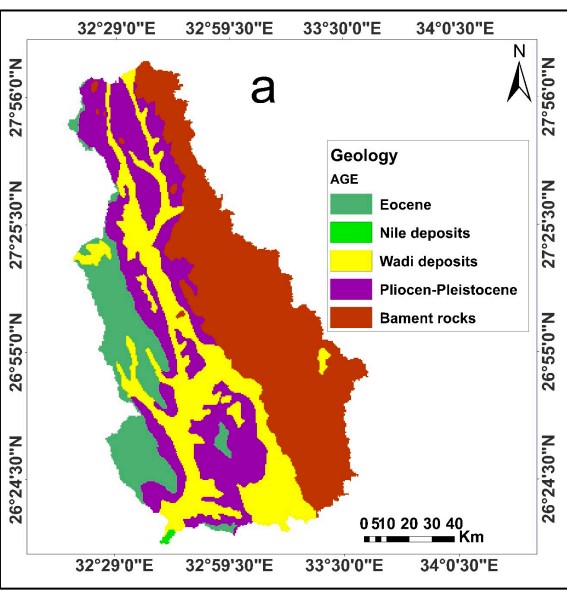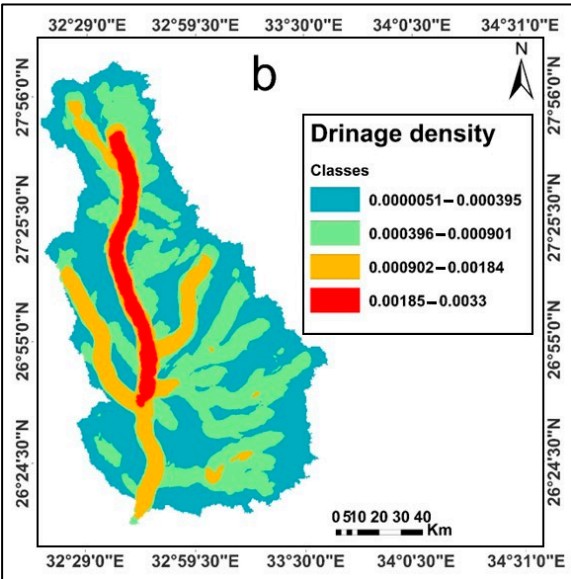

**Figure 4.** *Cont.*

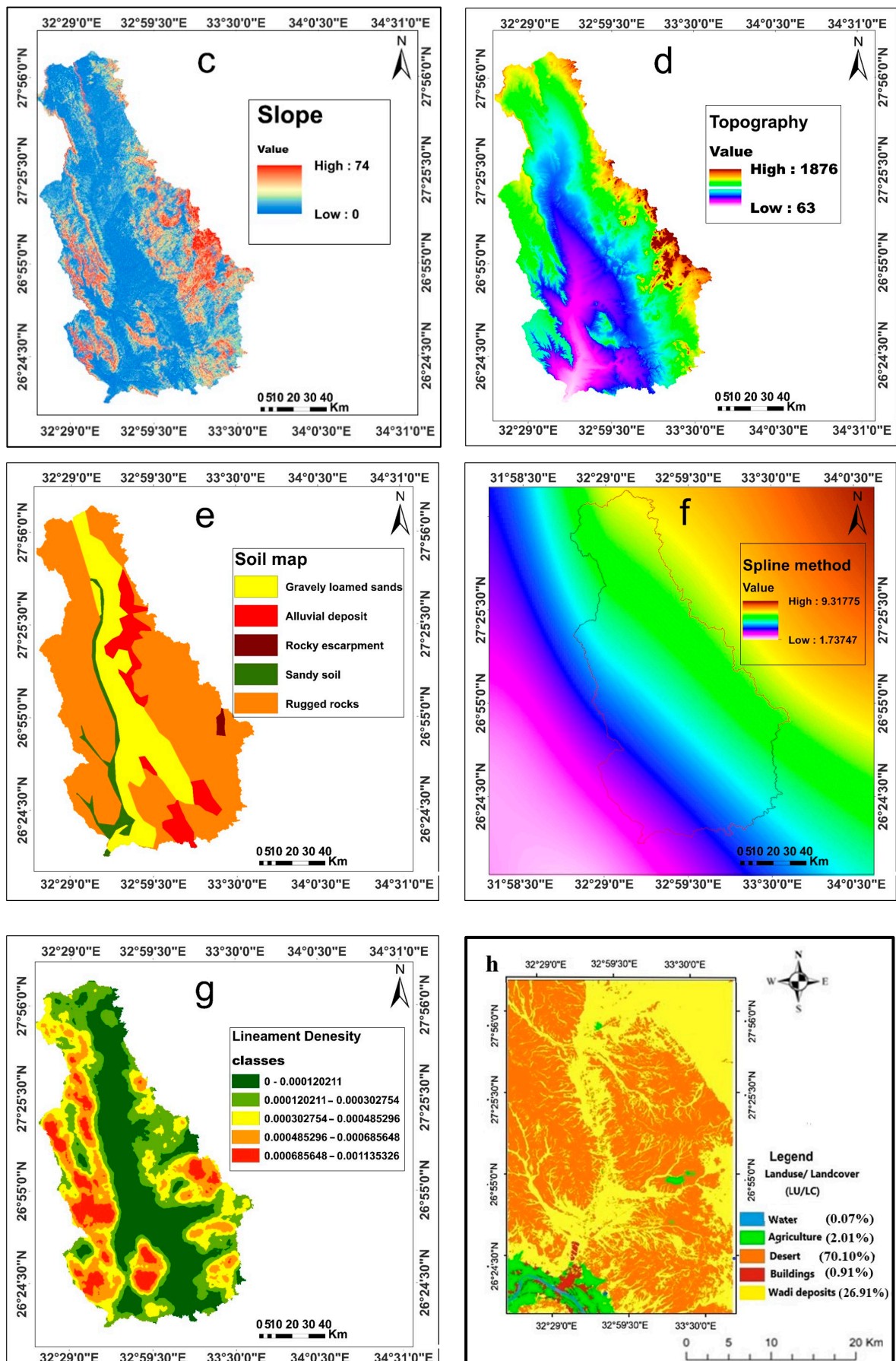

**Figure 4.** *Cont.*

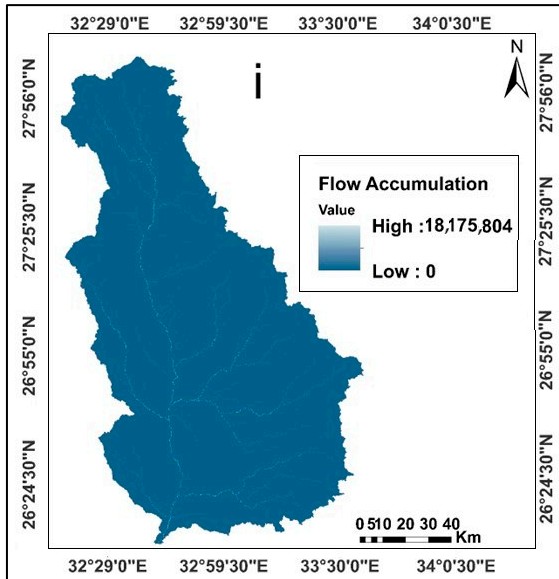 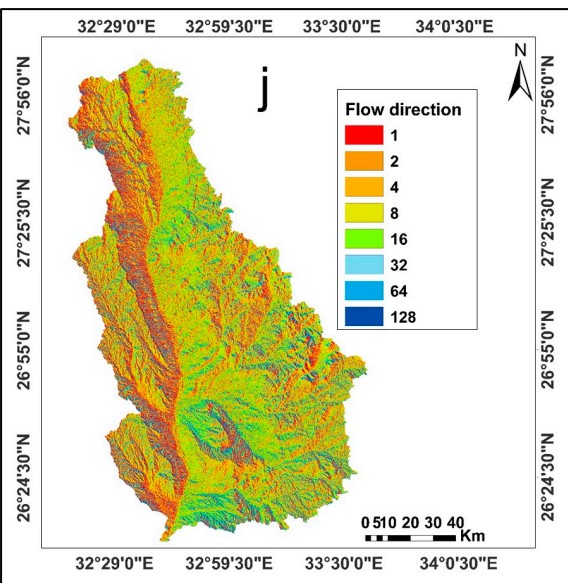

**Figure 4.** Thematic maps of (**a**) Geology; (**b**) Drainage density; (**c**) Slope, (**d**) Topography; (**e**) Soil; (**f**) Rainfall; (**g**) Lineament; (**h**) Land use/land cover (LULC); (**i**) Flow accumulation; and (**j**) Flow direction.

### 3.3. Drainage Density

The drainage density is an important part of how surface runoff works because it affects how strong floods are, how concentrated the water is, and even how much water is in a drainage basin. Drainage density is the total area of the drainage basin divided by the total length of the streams and rivers. It is a way to measure how well stream channels drain a watershed. High drainage density means more water runs off and less water soaks in (lower groundwater potential). The drainage density map of Wadi Qena Basin was constructed from STRM (DEM) using the spatial analysis tool, map algebra, and raster calculator in ARC GIS10.3 software. It was then categorized into four groups, numbered from 1 to 4, based on how they affected recharging and infiltration (Figure 4b).

### 3.4. Slope Map

The slope of the land controls the flow of surface water and accumulations. Flow can be faster down steep slopes, which leads to a shorter lag time and less recharge or infiltration. On the other hand, flow can be slower down gentle slopes, which leads to a longer lag time and more infiltration. A DEM was used to make a slope map of Wadi Qena Basin, and the slope layer was divided into five groups (Figure 4c). These groups are nearly level (0–4), gently sloping (4–10), moderately sloping (10–18), steep (18–30), and very steep or strongly steep (30–74). The almost flat to gently sloping land is good for groundwater recharge.

### 3.5. Topography Map

Areas with low topography often accumulate more rain, because the surface runoff is mostly of low speed, so there is a higher chance for more infiltration and high recharge ability into the GW aquifer in these areas. On the other hand, areas with high topography have high terranes and abrupt changes in the slopes, so the runoff gains more speed, allowing for less infiltration and GW aquifer recharge potentiality in these areas. The low-lying areas are the waterways and the Wadi bed which are filled with sediments from rivers. The high areas are made up of the Red Sea Mountains and some hills like G. Abu-Had. The topography map of Wadi Qena Basin was divided into five categories based on the STRM DEM and the ability to infiltrate and collect water (Figure 4d).

### 3.6. Soil Map

Based on the UNESCO classification (2006) (https://www.fao.org/soils-portal/data-hub/, accessed on 13 May 2023) and Landsat image (http://geo.arc.nasa.gov/sge/landsat/17.html, accessed on 13 May 2023), Wadi Qena Basin's soil map has five different types: alluvial deposit, gravelly loamed sands, sandy soil, rocky escarpment, and rugged rocks made up of limestone and basement rocks (Figure 4e). Founded on this map, the best places for water to enter are those with gravelly, sandy soil. Then, the map divided the soils into four groups based on their infiltration ability.

### 3.7. Rainfall Map

The rainfall data used in this study are an average of 29 years' values of annual rain from various stations around the valley, which were gathered from NASA's data (Table 1). Interpolating data was performed by the Spline method in ArcGIS, and the rainfall data for Wadi Qena Basin are classified into four groups (Figure 4f).

**Table 1.** Rainfall data from some stations for a long period (1990–2019).

| Station Name (North to South) | Longitude | Latitude | Elevation (m) | Avg. Rainfall (mm/Year) |
|---|---|---|---|---|
| Hurghada | 570,350.1 | 3,003,248 | 169.83 | 7.57 |
| Kossir | 629,964.3 | 2,890,730 | 116.06 | 5.9 |
| Minya | 276,940.2 | 3,108,145 | 132.24 | 8 |
| Assuit | 302,625.7 | 2,993,532 | 188.11 | 3.49 |
| Sohag | 370,560.3 | 2,942,790 | 286.88 | 2.27 |
| Luxor | 469,890.7 | 2,838,066 | 236.03 | 2.75 |
| South valley Uni. | 75,023.05 | 2,897,857 | 282.6 | 3.09 |

### 3.8. Lineament Map

Wadi Qena Basin has been subjected to several tectonic movements ranging from the Pre-Cambrian to the recent. Secondary porosity is caused by fault systems and fractures, which can control surface infiltration and predict the most important sites for groundwater abstraction. As a result, areas with high lineament density and major structures reveal abundant groundwater. Based on this information, Wadi Qena Basin was divided into four lineament density classes: very high, high, moderate, and low (Figure 4g).

### 3.9. Land Use/Land Cover Data (LULC)

Maps of global land use and land cover (LULC) (https://www.gislounge.com/2023-global-land-cover-data/, accessed on 13 May 2023) offer details on food security, hydrologic modelling, conservation planning, and the makeup of any given area on Earth. Sentinel-2 pixels with a resolution of 10 m by 10 m were used to create this map. The global land use/land cover data used in this study were obtained from Sentinel-2 (Figure 4h), with a 95% accuracy assessment rate using confusion matrix in ARC GIS software 10.1 (Table 2).

**Table 2.** Confusion matrix for accuracy assessment of LU/LC map, 2023.

| | LULC Classes | Desert | Wadi Deposits | Agriculture | Buildings | Water | Total | Users' Accuracy (%) |
|---|---|---|---|---|---|---|---|---|
| | | | | **Reference Data** | | | | |
| | Desert | **1912** | 48 | 21 | | | 1981 | 96.5% |
| | Wadi deposits | 20 | **1880** | 57 | 5 | 20 | 1982 | 94.8% |
| Classified Data | Agriculture | 56 | 19 | **1708** | | 8 | 1791 | 95.3% |
| | Buildings | 9 | 20 | 3 | **720** | 55 | 807 | 89% |
| | Water | 13 | 2 | 36 | 8 | **980** | 1039 | 94% |
| | Total | 2010 | 1969 | 1825 | 733 | 1063 | **7600** | |
| | Producer Accuracy (%) | 95% | 95.5% | 93% | 98% | 92% | | |

Overall Accuracy: 94.7, Kappa Coefficient: 0.92

### 3.10. Flow Accumulation

We created a flow accumulation map in ArcGIS using ASTER-GDEM data, and we reclassified it into five categories ranging from very low to very high (Table 3 and Figure 4i). Low accumulation values represent ridge tops, which have lower potential for groundwater infiltration due to steeper gradients and higher water flow. Higher accumulation levels, on the other hand, indicate valleys and stream systems, which increase the likelihood of groundwater due to strong infiltration.

**Table 3.** Thematic map weights and their capacity value for groundwater potential.

| Thematic Layer | Map Rank | Map Weight | Classes | Infiltration Range | Rank | Capability Value |
|---|---|---|---|---|---|---|
| Geology | 2 | 2/18 = 0.11 | Wadi deposits | Very high | 4 | 0.4 |
| | | | Pliocene | High | 3 | 0.3 |
| | | | Sedimentary succession | Moderate | 2 | 0.2 |
| | | | Basement rocks and Nile deposits | Low | 1 | 0.1 |
| Drainage density | 12 | 2/18 = 0.11 | 0.00000–0.0004 | Very high | 4 | 0.4 |
| | | | 0.0004–0.0009 | High | 3 | 0.3 |
| | | | 0.0009–0.00185 | Moderate | 2 | 0.2 |
| | | | 0.00185–0.00332 | Low | 1 | 0.1 |
| Slope | 3 | 3/18 = 0.166 | 0–4 | Very high | 5 | 0.33 |
| | | | 4–10 | High | 4 | 0.26 |
| | | | 10–18 | Moderate | 3 | 0.2 |
| | | | 18–30 | Low | 2 | 0.13 |
| | | | 30–74 | Very low | 1 | 0.06 |
| Topography | 2 | 2/18 = 0.11 | 65–200 | Very high | 5 | 0.33 |
| | | | 200–320 | High | 4 | 0.26 |
| | | | 320–460 | Moderate | 3 | 0.2 |
| | | | 460–950 | Low | 2 | 0.13 |
| | | | 950–1867 | Very low | 1 | 0.06 |
| Soil | 2 | 2/18 = 0.11 | Sandy soil and gravel | Very high | 4 | 0.4 |
| | | | Alluvial | High | 3 | 0.3 |
| | | | Rocky sands with lime | Moderate | 2 | 0.2 |
| | | | Hard rocks | Low | 1 | 0.1 |
| Rainfall | 1 | 1/18 = 0.05 | 2.9–4.2 | Low | 1 | 0.1 |
| | | | 4.2–5.1 | Moderate | 2 | 0.2 |
| | | | 5.1–6 | High | 3 | 0.3 |
| | | | 6–7.3 | Very high | 4 | 0.4 |
| Lineament density | 2 | 4/18 = 0.11 | 0–000147 | Low | 1 | 0.1 |
| | | | 0.000147–0.0004 | Moderate | 2 | 0.2 |
| | | | 0.0004–0.00059 | High | 3 | 0.3 |
| | | | 0.000598–0.001135 | Very high | 4 | 0.4 |
| LU/LC | 1 | 1/18 = 0.05 | Buildings | Very low | 1 | 0.1 |
| | | | Agriculture | Low | 2 | 0.2 |
| | | | Desert | Moderate | 3 | 0.3 |
| | | | Water | High | 4 | 0.4 |
| | | | Wadi deposits | Very high | 5 | 0.5 |
| Flow accumulation | 2 | 2/18 = 0.11 | <2 | Very high | 1 | 0.33 |
| | | | 2–3 | High | 2 | 0.26 |
| | | | 3–5 | Moderate | 3 | 0.2 |
| | | | 5–6 | Low | 4 | 0.13 |
| | | | >6 | Very low | 5 | 0.06 |
| Flow direction | 1 | 2/18 = 0.55 | <2 | Very high | 1 | 0.33 |
| | | | 2–8 | Very high | 2 | 0.26 |
| | | | 8–32 | High | 3 | 0.2 |
| | | | 32–64 | Moderate | 4 | 0.13 |
| | | | >64 | Low | 5 | 0.06 |

### 3.11. Flow Direction

The stream network is identified using a flow direction map, which provides the flow across a surface that will always be flowing in the sharpest downhill direction. The flow

direction map made from the ASTER-GDEM grid depicts how the river flows from the northeast to the southwest before arriving at the Nile (Figure 4j).

### 3.12. Groundwater Charateristics for Agriculture Use

Groundwater samples were gathered and evaluated for various parameters to ascertain their appropriateness for agricultural applications.

### 3.13. Physical Properties of the Groundwater Samples

Groundwater temperature is a consequential outcome of various heating mechanisms. It is widely acknowledged as a paramount determinant in the realm of groundwater quality management. It exerts a profound influence on nearly all physical phenomena and characteristics of water, encompassing chemical reactions and the presence of biological organisms within the aquatic ecosystem. The temperature of groundwater varies greatly with the geological structures, physiographical conditions, and the system of recharge. In the study area, groundwater temperature ranges between 22 and 28 °C.

The assessment of water pH holds significant importance in determining water quality. The pH level is influenced by the equilibrium of $CO_2^-$, $CO_3^-$, and $HCO_3^-$, which can be easily disrupted by alterations in carbon dioxide content. In the investigated region, the groundwater exhibited a pH range of 7.5 to 8, indicating an alkaline nature. These pH values are deemed favorable and contribute positively to the nourishment of nutrients in the soil and the efficacy of added fertilizer elements.

### 3.14. Chemical Properties and Composition of the Groundwater

The measured electric conductivity values ranged between 3.6 to 13.8 mS cm$^{-1}$. Total dissolved solid (TDS) ranges from 2304 to 8832 ppm. The distribution map of TDS in the studied area exhibits an upward trend towards the southern region of the Wadi. Furthermore, the salinity levels in the Quaternary aquifer surpass those observed in the Nubian aquifer. This rise in salinity towards the south could potentially be attributed to the prevailing direction of groundwater flow, indicating the influence of leachate effects resulting from water-bearing fertilization. Additionally, it is plausible that this phenomenon signifies a consistent replenishment of groundwater through precipitation.

In the studied area, the range of sodium concentrations varied from 690 ppm to 1265 ppm. It is observed that the sodium content generally exhibits an upward trend from the northern region to the southern region within the Quaternary aquifer. Conversely, within the Nubian aquifer, the sodium content demonstrates an increasing pattern from the southern region to the northern region.

Calcium content ranged between 40 and 100 ppm. Generally, the calcium content typically exhibits an ascending trend from the northern regions towards the southern areas. The magnesium content in the studied area ranged between 14.64 and 80 ppm. The magnesium concentration increases from north to the south in the Quaternary aquifer with the flow direction, while it increases from the south to the north for the Nubian aquifer (in the reverse direction).

The bicarbonate ion ($HCO_3^-$) concentrations in the groundwater in the study area ranged between 24.4 and 91.5 ppm. Chloride ($Cl^-$) concentration in the studied area ranged between 887.5 to 1775 ppm. The concentration of sulfate ions ($SO_4^{2-}$) within the study area varied from 240 to 576 ppm. It is observed that the sulfate content of QAS generally exhibits an increasing trend towards the southern region of the study area. Conversely, the sulfate content of NSAS shows an increasing trend towards the northern region.

In the present study, the groundwater suitability for irrigation purposes is calculated and classified according to sodium adsorption ratio (SAR). The SAR values ranged between 23 and 32 in the studied groundwater samples. Based on the SAR index, most of the water samples are unsuitable for irrigation. The water quality of the southern part of the area has a high sodium content, which is classified as fair or suitable for agriculture irrigation. The potential risks associated with water utilization may result in detrimental consequences that

can be expected in a majority of soil types. In order to mitigate these effects, the application of amendments such as gypsum becomes imperative to facilitate the exchange of sodium ions. This process necessitates the presence of adequate drainage and the utilization of chemical amendments. Conversely, the northern region of the Wadi is distinguished by water quality characterized by an exceedingly high sodium content, rendering it unsuitable for agricultural irrigation. Consequently, the hazards associated with water use generally render it unsatisfactory for irrigation purposes, particularly due to its poor quality and the requirement for low-salinity water.

## 4. Discussion

### 4.1. Groundwater Prospect Map

After assigning a weight factor to thematic layers, GIS techniques enabled the creation of a groundwater potential map. Groundwater potential zones are created by combining multiple GIS layers. This is referred to as multi-criteria evaluation [3]. All thematic maps were ranked into different classes and ranked from 1 to 5 based on the infiltration properties and weighted according to their relative importance to control groundwater potential, with 1 being the least and 5 being the most important for recharge potentiality (Figure 5). The equation's final prospect map was classified into five categories of groundwater prospects: very high, high, moderate, low, and very low zones (Figure 6). Wadi deposits and deeply weathered metamorphic rocks are represented by the excellent and high sites. However, the limestone plateau and younger rocks are distinguished by moderate and low prospect zones. The prospect map produced was compared to the findings of previous studies such as [36,40] and field observation, and the proposed model was valid.

### 4.2. Field Observation

The groundwater recharge potential map of the study area was validated using field observation. We gathered data on the existing wells (Table 4) to validate the combined RS and GIS results. The location of the wells and the water depth were the main goals of the field trip. The data comprised surface and near-surface geology, topographic setting, and soil type. These wells were drilled through the valley floor of the Quaternary aquifer, and their intercalation depths range from 4 to 120 m (Figure 7a,b). These wells' bottoms are near barrier rocks like clay and silt beds, which prevent deeper water infiltration. It has been shown that the yield of these wells grows southerly and ranges from 10 to 50 m$^3$ per day. The salinity ranges from 750 to 2000 ppm, and the quality is good or satisfactory [28]. The surface soil succession is mainly made up of silty sand, gravel sand, and sandy gravel (Figure 7c). The valley bed's succession through it is fragmented, serving as a pathway for water infiltration, and it coincides with the excellent potential area in the GW recharge potential map. Table 4 indicates clearly that the depth of the water table in the wells in Wadi Qena Basin changes. This might be ascribed to differences in topography, water well depths, the general tilt of Wadi Qena Basin, and the distance from the Nile River aquifer. As a result, local farmers have developed the southern part of Wadi Qena Basin (Figure 7d) and use shallow groundwater of high quality to irrigate several crops, and it coincides with the high potential area in the GW recharge potential map (Figure 6).

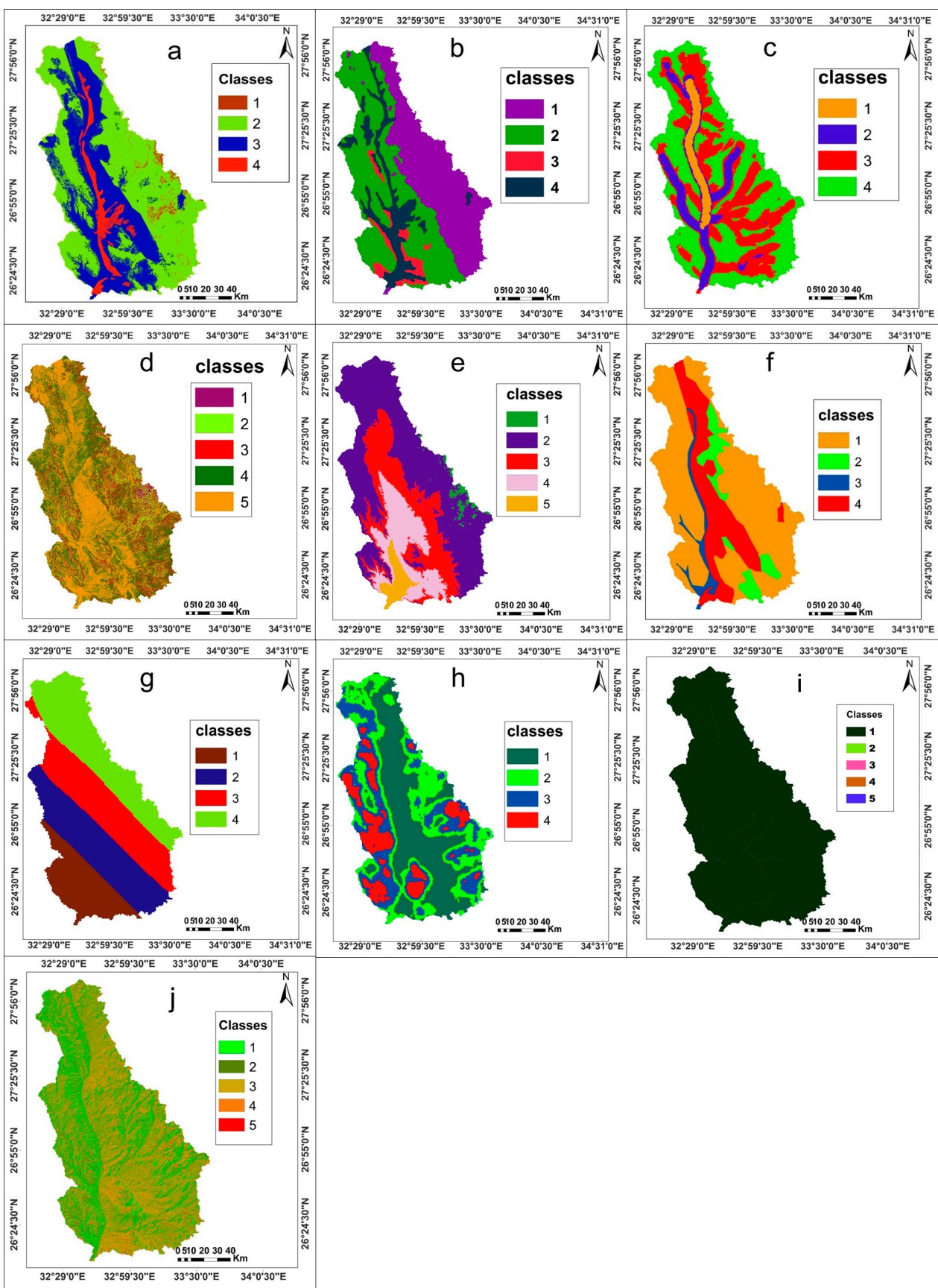

**Figure 5.** The classes of all thematic maps. (**a**) Classes of geology;(**b**) Classified drainage density; (**c**) Classes of slope; (**d**) Classes of topography; (**e**) Classes of soil; (**f**) Classes of rainfall; (**g**) Classes of Lineament density; (**h**) Classes of LULC; (**i**) Flow accumulation; and (**j**) Flow direction map.

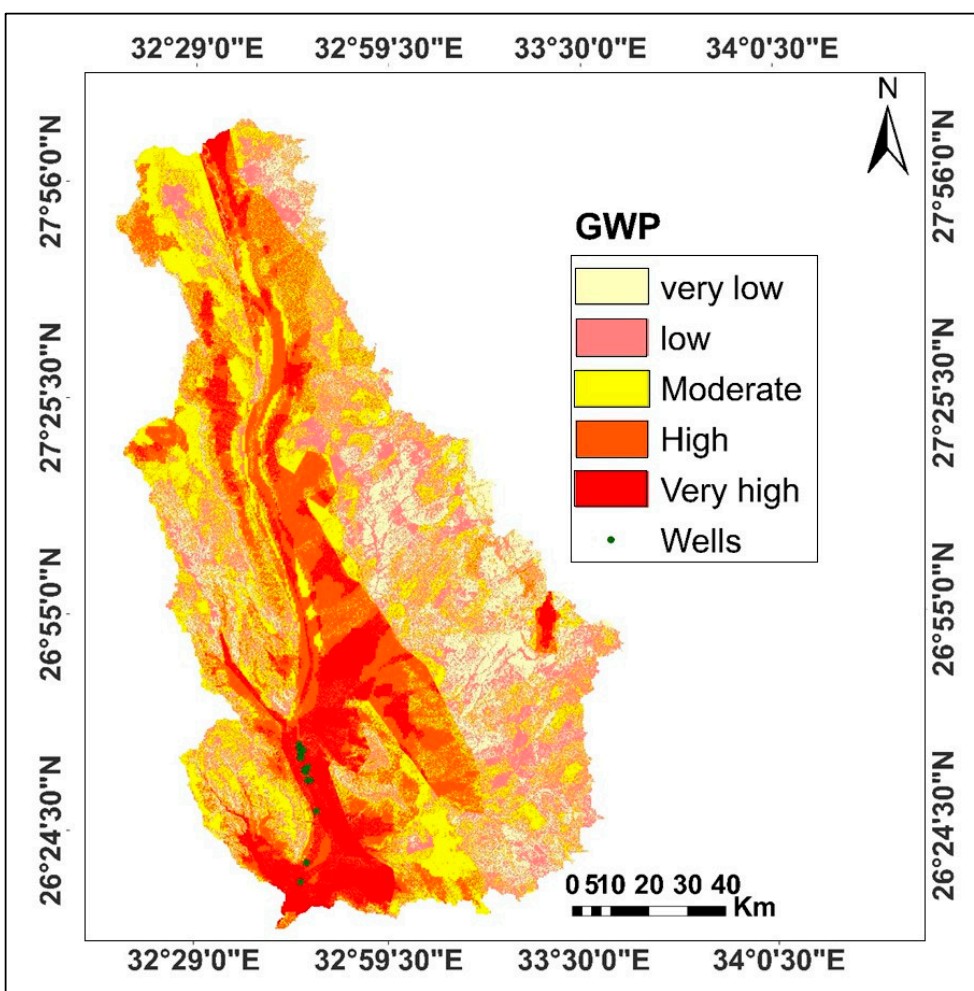

**Figure 6.** Groundwater recharge potential map.

**Table 4.** Geographic location and depths of wells at Wadi Qena Basin area.

| Well No. | Latitude | Longitude | Well Depth (m) | Topographic Elevation (a.s.l) | Water Table (m) |
|---|---|---|---|---|---|
| 1 | 26.260 | 32.760 | 30 | 100 | 70 |
| 2 | 26.290 | 32.767 | 30 | 100 | 70 |
| 3 | 26.294 | 32.768 | 36 | 105 | 69 |
| 4 | 26.296 | 32.768 | 35 | 110 | 70 |
| 5 | 26.331 | 32.780 | 50 | 115 | 60 |
| 6 | 26.670 | 32.786 | 8 | 125 | 107 |
| 7 | 26.315 | 32.745 | 4 | 195 | 121 |
| 8 | 26.386 | 32.780 | 120 | 110 | 75 |
| 9 | 26.400 | 32.798 | 5 | 130 | 105 |
| 10 | 26.523 | 32.802 | 15 | 135 | 115 |
| 11 | 26.266 | 32.774 | 15 | 165 | 120 |
| 12 | 26.78 | 32.783 | 80 | 115 | 110 |
| 13 | 26.618 | 32.788 | 5 | 200 | 36 |
| 14 | 26.810 | 33.463 | 12 | 647 | 584 |
| 15 | 26.265 | 33.458 | 16 | 705 | 109 |
| 16 | 26.404 | 32.788 | 10 | 115 | 130 |
| 17 | 26.257 | 32.761 | 12 | 600 | 75 |

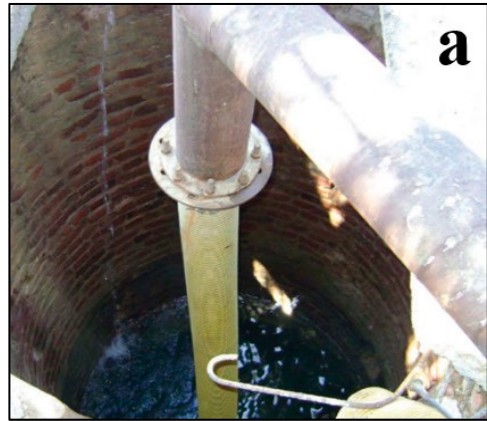
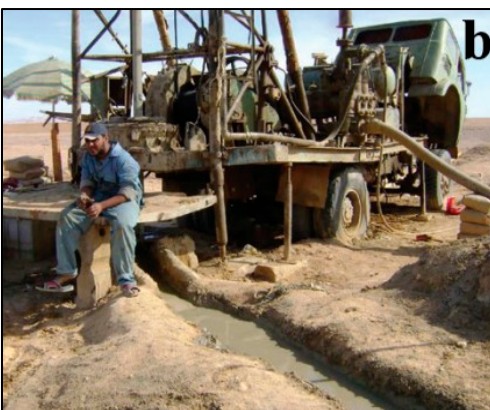
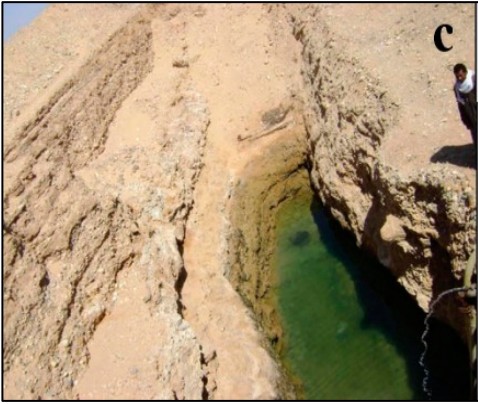
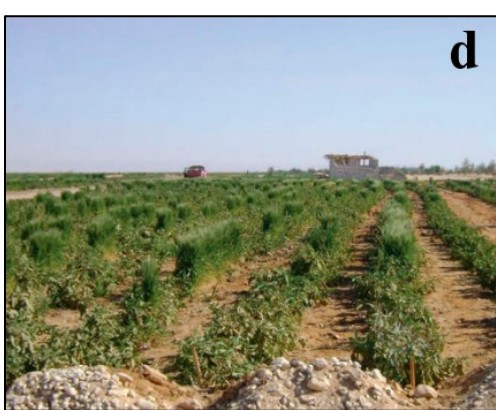

**Figure 7.** Field photographs of the study area: (**a**) a highly productive well with 4 m depth; (**b**) well drilling at central Wadi Qena Basin to a depth of 120 m; (**c**) excavated water well at inter-bedded sand and gravel, bounded by shale; (**d**) corn and tomato crops at the southern part of Wadi Qena Basin.

### 4.3. Soil Studies and Field Survey

The samples were collected from the well-drained landform units, which from the field survey can be made available for land reclamations such as Wadi floor (WF), low-elevated sand sheet (LSS), high-elevated sand sheet (HSS), Bajada (B), Piedmont (P), and tableland (TL). With a few exceptions that could be ascribed to elevation, the analyzed units of the research area have comparable morphological characteristics, which is consistent with studies described by [71].

The highest gravel content was found in TL profiles, which had a range of 7 to 45%. Sand-like coarse texture was a defining characteristic of LSS, HSS, and TL units, while the WF, B, and P units contained loamy sand. When compared to other soils, WF soils have a larger content of fine soil particles, which can be attributed to the deposition of these particles from higher topographic sites. The bulk density of the soil varied between 1.33 and 1.46 g/cm$^3$. The findings showed that the soils have poor water supply capacity, which can be related to their coarse texture and low levels of organic carbon. Due to their finer soil texture and higher levels of organic matter, WF and B soils displayed the maximum amount of water that was readily available.

In comparison to other soils, the calcium carbonate concentration of the WF, B, and P soil profiles was higher. Additionally, the examined soils showed increased calcium carbonate concentration in the upper portions of their soil profiles, which indicates that the nearby calcareous desert zone has an impact on them due to wind activity. Additionally, the common flash floods that occur along the eastern higher relief limestone plateau may potentially be a factor in the deposition of calcareous disintegration products.

At the subgroup level, two soil orders, *Entisols* and *Aridisols*, were identified, and keyed out as *Typic Torripsamments* and *Typic Quartzipsamments* and *Typic Haplocalcids*, respectively.

The soils were slightly to strongly alkaline in soil response (pH 7.5 to 8.6), moderately deep to deep in depth, well-drained, and mildly to moderately saline (EC 3.6 to 16.1 dS m$^{-1}$). The soils had low levels of calcium carbonates (1.1–19.7%), low levels of CEC (1.3–11.4 cmol$^{+}$), and low levels of organic matter (0.03–0.53%). The findings showed that the soils had low to moderate levels of accessible N (1–14 kg ha$^{-1}$), P (1–7 kg ha$^{-1}$), and K (87–435 kg ha$^{-1}$). Additionally, the soils have few readily available micronutrients. These results are in agreement with the findings of previous works [71].

The current analysis revealed that the soil in Wadi Qena had poor physical and chemical characteristics. As a result of its coarse texture, low OC and CEC, and alkaline soil pH, the soil possessed a low capacity for providing nutrients and water. These results are confirm those of [71,72].

From Figure 8, the agricultural suitability of the soils in the study area spanned from poor to fair. Additionally, the soils exhibited a range of suitability for growing various crops, including fields, vegetables, forage crops, and fruits. Wheat, barley, sunflower, date palm, olive, fig, grape, alfalfa, sorghum, and beet were identified as the most suitable crops for cultivation in the study area. The primary limiting factors that influence land capability and crop suitability include coarse soil texture, soil salinity, sodicity, organic matter content, and nutrient availability. The Wadi Qena region has been identified as a crucial area for agricultural expansion, as per the land assessment conducted. It has been established that this region holds immense potential for reclamation and is a significant target for sustainable agriculture [73,74].

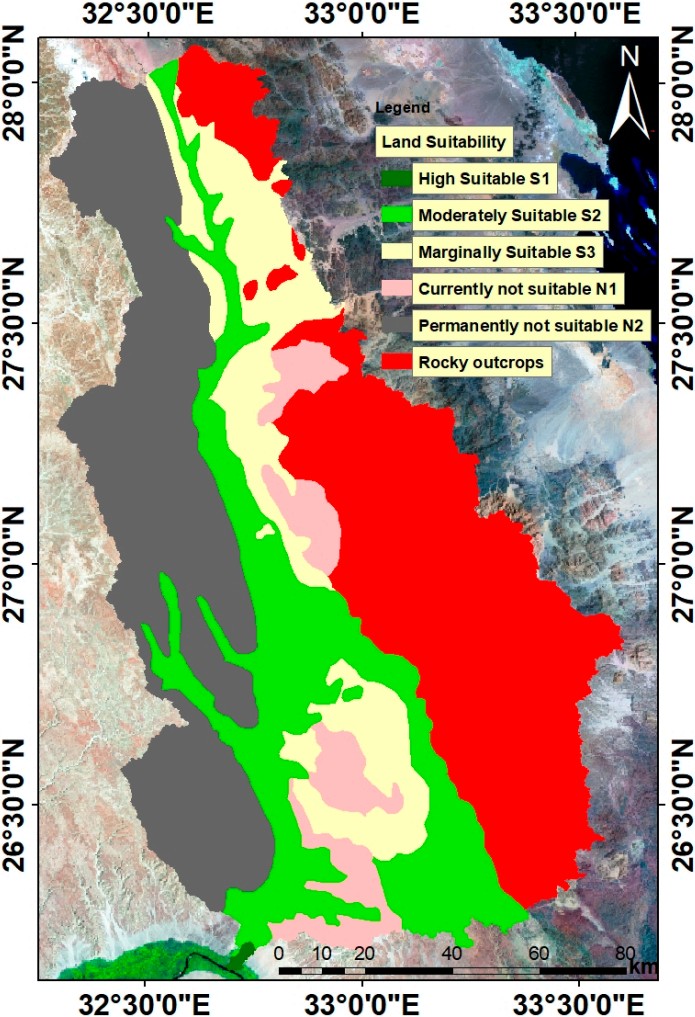

**Figure 8.** Land suitability in Wadi Qena.

In this study, the assessment of water quality for irrigation demand was conducted in order to meet the requirements of crop irrigation water quality. The recommendations from the FAO 29 guideline for irrigation water quality and an iterative survey of irrigation experts were utilized for this purpose. It is crucial to monitor irrigation waters in areas where irrigation is utilized as an end use. It is advisable to adjust the crop pattern before the commencement of each growing season to ensure sustainable irrigation. The findings of the study indicated a range of water quality from very poor to satisfactory along the Wadi. The quality of water is influenced by seasonal patterns and exhibits a cyclical nature. To further enhance the water quality, it is recommended to explore alternative indices that can serve as a basis for an irrigation index. Additionally, this monitoring process will aid in revising the crop pattern for irrigated land to comply with the water quality of the respective year. Importantly, the adoption of the available methods will enhance the accuracy of the assessment, leading to more reliable results.

In relation to agricultural management aspects that will be necessary in a development that is unlikely to be sustainable in the long run due to limited recharge from a minimal amount of rainfall, there are several additional issues that could be addressed. These include the management of access and entitlements to groundwater, with the aim of ensuring that the focus remains on maintaining the sustainability of this vital resource. It is crucial to avoid excessive pumping and implement effective control measures in the designated area. Furthermore, the significance of managing fertilizer usage in soils that are known to have low natural fertility should be emphasized in order to prevent any potential pollution of the aquifer. Additionally, exploring the potential for desalination of brackish groundwater through the utilization of reverse osmosis, powered by solar photovoltaic electricity generation in an area with high levels of radiation, could provide an opportunity for cultivating high-value intensive crops.

## 5. Conclusions

The Wadi Qena Basin, situated in Egypt, is recognized as one of the largest wadis in the country. It is positioned in the northeastern region of Qena, spanning between latitudes $26°00'$ and $28°20'$ N and longitudes $32°00'$ and $33°00'$ E, covering an approximate area of $16,000$ km$^2$. The construction of a new road connecting upper Egypt governorates and the Red Sea has led to various forms of agricultural development in the Wadi. In order to assess the basin, the current study employed an assessment software package called Arc GIS 10.3, along with STRM (DEM) data with a resolution of 30 m. These data were utilized to generate maps of topography, slope, aspect, contour, drainage network, flow accumulation, and flow direction within the Wadi Qena Basin. Additionally, satellite images such as Shuttle Radar Topography Mission (STRM) and Landsat data were processed and combined to predict the most suitable zones for groundwater recharge. The study employed GIS spatial analysis to integrate multiple layers of data, including topography, geology, slope, drainage density, soil, rainfall, and lineament maps. This approach successfully identified the optimal locations for groundwater recharge potential. By utilizing the GIS re-classify tool, the potential map was divided into five zones, ranging from very high to very low groundwater recharge potential. This classification is a valuable contribution to water management in the Wadi Qena Basin. The accuracy of the final map was confirmed through the use of drilled wells, which revealed several underutilized locations with groundwater potential. Planners and decision-makers can utilize the groundwater potential zone (GWPZ) map to guide future development in the area. The study emphasizes the critical importance of careful groundwater potential management and regular monitoring to preserve the current state of groundwater in the Wadi Qena Basin. Furthermore, the integrative approach employed in this study can be applied to other basins worldwide, showcasing the spatial distribution of groundwater recharge potential in Wadi Qena and serving as a significant input for sustainable agriculture.

**Author Contributions:** Conceptualization, H.A.M., A.E.-H.A.F, A.A.M., A.S. and M.A.E.A.; methodology, H.A.M., A.E.-H.A.F, A.A.M., P.D. and M.A.E.A.; software, H.A.M., A.E.-H.A.F, A.A.M. and M.A.E.A.; validation, H.A.M., A.E.-H.A.F, A.A.M., P.D. and M.A.E.A.; formal analysis, H.A.M., A.E.-H.A.F, A.A.M. and M.A.E.A.; investigation, H.A.M., A.E.-H.A.F, A.A.M. and M.A.E.A.; resources, H.A.M., A.E.-H.A.F, A.A.M., P.D., A.S. and M.A.E.A.; data curation, H.A.M., A.E.-H.A.F, A.A.M., A.S. and M.A.E.A.; writing—original draft preparation, H.A.M., A.E.-H.A.F, A.A.M., P.D., A.S. and M.A.E.A.; writing—review and editing, H.A.M., A.E.-H.A.F, A.A.M., P.D., A.S. and M.A.E.A.; visualization, H.A.M., A.E.-H.A.F. and M.A.E.A.; supervision, H.A.M., A.E.-H.A.F, A.S., and M.A.E.A.; project administration, H.A.M., A.E.-H.A.F, P.D., A.S. and M.A.E.A.; funding acquisition, H.A.M., A.E.-H.A.F, A.A.M., P.D., A.S. and M.A.E.A. All authors have read and agreed to the published version of the manuscript.

**Funding:** This research received no external funding.

**Data Availability Statement:** The data used to support the results of this research are available on request from the corresponding authors.

**Acknowledgments:** The manuscript presents a participation between the scientific Institutions/Faculties in two countries (Egypt and Italy), and in particular, the authors are grateful for their support in carrying out the work to: (1) National Authority for Remote Sensing and Space Sciences (NARSS), Cairo 11769, Egypt; (2) Faculty of Science, Assiut University, Egypt; (3) Scuola di Scienze Agrarie, Forestali, Alimentari ed Ambientali (SAFE) SOILSCOSOF Unibas (Italy).

**Conflicts of Interest:** The authors declare no conflict of interest.

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
