# Peer review of "Groundwater Recharge Potentiality Mapping in Wadi Qena, Eastern Desert Basins of Egypt for Sustainable Agriculture Base Using Geomatics Approaches"

_hydrology, doi:10.3390/hydrology10120237_

Round 1
Reviewer 1 Report
Comments and Suggestions for Authors
This paper aims to locate groundwater recharge potential zones 20 (GWPZs) in Wadi Qena Basin, Eastern Desert of Egypt using Multi-criteria evaluation (MCE), to explore the opportunities for land reclamation for intensive irrigated agriculture based on access to groundwater from shallow aquifers. The paper has a carefully defined , excellent description of the hydrogeology of the area and its associated aquifers and water quality, accompanied by excellent figures The paper recommends that groundwater extractions in the Wadi Qena basin take place near the identified, deeply buried faults, but points out that most of the groundwater is fossil water, and that while some recharge is evident in the Nubian aquifer, it is hardly significant (not surprising in an area limited to 2-8mm annual rainfall) while upward recharge from the Basement complex faults are not expected to be a significant source of recharge. The paper identifies a recharge area located in the northeastern region of the study area with groundwater flow towards the south-west. Flow in the quaternary aquifer is also to the south, where it has potential for domestic and agricultural purposes, but with increasing salinity.
Although the paper is generally excellent as far as it goes and describes valuable investigations, the authors might consider further discussing the agricultural management aspects that will be required in a development that is unlikely to be sustainable in the long term due to limited recharge from a very small rainfall. Issues that could be further mentioned include management of access and entitlements to the groundwater to ensure orientation towards maintaining sustainability of the resource (noting observations that some excessive pumping is already taking place) , any scope to develop stormwater run-off impoundments for possible aquifer recharge via percolation basins (but noting evaporation losses) or recharge wells, the importance of managing fertiliser use in soils stated to be of low natural fertility to preclude aquifer pollution, and scope for desalination of brackish groundwater for high value intensive crops using reverse osmosis driven by solar photovoltaic electricity generation in an area with high radiation levels.
The following specific comments are made:-
Line 50 – It may be useful to define the meaning of “wadi” those readers not familiar with the term.
Lines 78, 135, 140, 140, 360, there are references to “red sea hills”, “Red Sea mountains”, “red sea peaks” and the “Red Sea”. The authors might like to consider the need for consistent capitalisation and any clarification between the terms “hills”. peaks and “Mountains”. It is suggested that the Red Sea Mountains be identified on Figure 1.
Line 78 =- “…70% of the testes had infiltration rates of less than 2 m/day,…”. Check. Presumably you mean “tests”:, not “testes” which are male sex glands.
Line 204 – The authors might consider deleting the term “:cutting-edge” from the text – an over-used and clichéd term not really appropriate for a scientific paper.
Line 210 – What is meant by the expression “contrive entrance”?
Line 210-212 – review construction of sentence “Also….change” as meaning is confusing.
Line 214-215 – Insert “were” before “used”.
Line 221-222 – Insert “was” before “created”. (These two sets of lines lack a verb.)
Figure 3 – Is a reference available for the geological map “CONOCO 1987”?
Table 1 – misspelling of “year” in right hand column.
Table 4 and its caption have no units. Presumably the various measures are in metres, but this should be stated.
Line 543 – What is “bear”? If it is a local name for a crop, please define. Do you mean “beet”?
Comments on the Quality of English Language
See queries raised above.
Author Response
Dear colleague,
Thank you for taking the time to review our manuscript. We hope that we have been able, with the changes made, to improve the paper. Your suggestions were very helpful, and on behalf of all the authors I thank you.
Best regards
Prof. Antonio Scopa

Reviewer 2 Report
Comments and Suggestions for Authors
The manuscript describes a multi-criteria evaluation of the recharge potential of the Wadi Qena Basin aquifer, using remote sensing, GIS and analytical hierarchical classification. It then attempts to calibrate some of the observations with some field measurements, and then highlights the potential of the findings for agriculture.
The MS however is somehow confusing and does not follow a strict scientific logic. The long introduction is mainly devoted to describing the study area, and most of this section should go to the ¨study area¨ section. I stronly recommend to insert more background to the specific topic you´re dealing with, such as application in other regions, shortcomings of the methods and possible handicaps of using certain fixed hierarchies, and how some could be reconsidered as for their significance.
From section 3.14, the authors jump into hydrochemical and soil analyses, but it is not clear how this relates to the GIS and RS findings described in previous sections. It appears as if the recharges topic is not fully related to the field-analytical topic. This potential relationship and its investigation approach, should in turn be explained in detail in the introduction, making it clear what one should expect from the study.
With clearer hypothesis and objectives stated at the beginning, the conclusions would certainly benefit.
The conclusion as it stands, is too long with not much actual conclusions. Only at the end of the section it reads: ¨The study concluded that careful groundwater potential management and regular monitoring are critically important to preserve the current state of groundwater in the studied area. This study's integrative approach, which may be used in other basins throughout the world, demonstrates the spatial distribution of groundwater recharge potentiality in Wadi Qena. Meanwhile it is an important input 589 for sustainable agriculture use.¨
I strongly suggest to reconsider and rewrite the conclusions, trying to give an answer to the objectve(s) which in turn should be written much clearer at the end of introduction.
Iy is an interesting and relevant work, that could be published onces the story between GIS/RS and water/soils quality is better articulated.
Figures have different fonts and sizes, and should be improved in quality, and be homogenous in their display. The water table map, lacks quality.
Line 63: mentions AHP before it is explained
Line 88: testes (tested)
Line 89: repeats infiltration
etc.
Comments on the Quality of English Language
Only a few flaws should be reviewed and corrected.
Author Response

(The authors gave the same response as above.)
